# Simulation of Ultrasound RF Signals Backscattered from a 3D Model of Pulsating Artery Surrounded by Tissue

**DOI:** 10.3390/diagnostics12020232

**Published:** 2022-01-18

**Authors:** Monika Makūnaitė, Rytis Jurkonis, Arūnas Lukoševičius, Mindaugas Baranauskas

**Affiliations:** Biomedical Engineering Institute, Kaunas University of Technology, K. Baršausko Str. 59-455, LT-51423 Kaunas, Lithuania; rytis.jurkonis@ktu.lt (R.J.); arunas.lukosevicius@ktu.lt (A.L.); m.baranauskas@ktu.lt (M.B.)

**Keywords:** carotid artery, ultrasound, scatterers, motion simulation, Field II

## Abstract

Arterial stiffness is an independent predictor of cardiovascular events. The motion of arterial tissues during the cardiac cycle is important as a mechanical deformation representing vessel elasticity and is related to arterial stiffness. In addition, arterial pulsation is the main source of endogenous tissue micro-motions currently being studied for tissue elastography. Methods based on artery motion detection are not applied in clinical practice these days, because they must be carefully investigated in silico and in vitro before wide usage in vivo. The purpose of this paper is to propose a dynamic 3D artery model capable of reproducing the biomechanical behavior of human blood vessels surrounded by elastic tissue for endogenous deformation elastography developments and feasibility studies. The framework is based on a 3D model of a pulsating artery surrounded by tissue and simulation of linear scanning by Field II software to generate realistic dynamic RF signals and B-mode ultrasound image sequential data. The model is defined by a spatial distribution of motions, having patient-specific slopes of radial and longitudinal motion components of the artery wall and surrounding tissues. It allows for simulating the quantified mechanical micro-motions in the volume of the model. Acceptable simulation errors calculated between modeled motion patterns and those estimated from simulated RF signals and B-scan images show that this approach is suitable for the development and validation of elastography algorithms based on motion detection.

## 1. Introduction

Cardiovascular diseases (CVDs) are the main cause of human death worldwide [1]. Stroke, one of the several possible outcomes of CVD, represents the second cause of mortality and the sixth cause of permanent disability [2,3]. Prediction of cardiovascular events towards efficient patients’ treatment is, therefore, a major public health issue [4,5,6,7,8]. Atherosclerosis, the underlying pathological mechanism, is a chronic and systemic vascular disease often referred to as a “silent killer”, because its progression and evolution are tremendously complex and rather unpredictable, leading to an exceedingly high number of fatal events occurring without any precautionary sign [9]. Overall, atherosclerotic arterial affection is not noticeable in the long term, but general signs of this disease develop only after the onset of complications: thickening of the intima–media complex, narrowing of the lumen or its thrombosis, and/or loss of elasticity [10].

Arterial stiffness is an independent predictor of cardiovascular events, and it is analyzed to assess cardiovascular risk [11,12,13]. Within the context of CVD risk assessment, the motion of arterial tissues during the cardiac cycle is important, as mechanical deformation represents vessel elasticity and is putatively inversely related to arterial stiffness, an independent indicator of all-cause mortality and morbidity [14]. Arterial tissue motion during the cardiac cycle consists of radial (artery wall motion perpendicular to blood flow), longitudinal (intima–media complex motion in parallel with blood flow) motions, and circumferential strain. Using several techniques, the motion of the arterial tissue from sequences of ultrasound (US) radiofrequency (RF) signals or images has been analyzed [15].

Ultrasound elastography (USE), which has been developing since 1990, is a non-invasive method that measures the stiffness or elasticity of tissues by measuring tissue deformation or movement. Elastography techniques rely on altered soft tissue elasticity to detect pathological or physiological processes. For example, many tumors have different mechanical properties than the surrounding healthy tissues. In liver fibrosis, the liver becomes stiffer than healthy tissues. Thus, elastography allows the differentiation of healthy and damaged tissue for diagnostic purposes [16,17].

Currently, available USE techniques can be classified into two major groups according to the measured physical quantity: (1) strain imaging and (2) shear wave imaging. In the first approach, the strain is measured when the stress is applied to the tissue and the qualitative Young’s modulus is evaluated. To this group, strain elastography (SE) and acoustic radiation force impulse (ARFI) strain imaging methods are assigned. In shear wave imaging, dynamic stress is applied to tissue, the shear wave speed is measured, and quantitative Young’s modulus is computed. One-dimensional transient elastography (1D-TE), point shear wave elastography (pSWE), and two-dimensional shear wave elastography (2D-SWE) are currently available for shear wave imaging [16]. Summing up, all these methods use an external source of stress. Nevertheless, an internal source of deformation, or so-called endogenous motion in the tissue induced by heartbeat and vascular activity, could be used in elasticity measurement [18].

Methods based on endogenous motion detection are not applied in clinical practice these days. First, these methods should have three “in” steps before wide usage: (1) approach justification in silico to understand, develop, and verify the method; (2) testing in vitro to optimize, doublecheck, and validate pilot results; finally, (3) certification in vivo to fine-tune and confirm the reliability of the proposed method. Due to the fact of this, before in vitro and in vivo, we could use computational ultrasound imaging to simulate US RF signals or B-scan images from measurements using algorithms that are dedicated for this purpose.

The motion of the carotid artery wall has been modeled by many authors. Gemignani et al. [19] simulated radial motion as a sawtooth function, while Solomon et al. [20] simulated it as a triangular function. As reported by Deng et al. [21], they proposed a geometric model of a common carotid artery (CCA) where the dynamic scatterers model is constructed by moving the positions of the scatterers according to the synthesized pulse wave (PW). The PW is estimated from the in vivo RF signals of the vessel wall using a 1D normalized cross-correlation (NCC) algorithm based on echo tracking. For the motion in the radial direction, Hu et al. [22] proposed an isotropous pulse model with elasticity to describe the relationship between radial displacement and blood pressure. However, neither of the previous studies consider the longitudinal motion of the artery together with radial and longitudinal motions slope deep into the tissue. This was performed by Stoitsis et al. [23] by simulating the radial motion as a pulse function that had parameter values that were determined based on experiments with in vivo data. The longitudinal motion was modeled as a sinusoidal function because there is no conclusive information about the mathematical modeling of this motion. Based on a similar in vivo examination, the spatial dependence of radial and longitudinal motions, which are independent of the axial position, was also incorporated.

In this study, a combination of the Field II program and in vivo arterial wall motions (i.e., radial and longitudinal) was used to generate realistic dynamic RF signals and B-mode ultrasound image sequential data. The latter may be used for the development and validation of algorithms for the processing of RF signals or B-mode images (intima–media complex segmentation, micro-motion tracking for characterization of tissue elasticity, etc.).

The paper aimed to propose a dynamic 3D artery model capable of reproducing realistic dynamic RF ultrasound signals backscattered from a pulsating human blood vessel surrounded by elastic tissue for 3D deformation elastography simulations, development, and verification of elastography algorithms, 3D elastography experiments, and feasibility studies.

## 2. Materials and Methods

The materials and methods section is organized into four sub-sections and each of them stands for the corresponding four stages of the proposed algorithm as shown in Figure 1. The purpose of this algorithm flow is to show the input and output parameters and highlight the ability of the simulation algorithm to modify the input parameter in order to adapt it to a specific simulation task.

In the first stage, the artery is modeled by the static scatterer’s distribution according to the vessel’s geometry and position. Subsequently, a static artery model is modified to a dynamic one by giving the radial and longitudinal motions to 3D distributed scatterers of the artery (Stage 2). Pulsation waveforms required to input Stage 2 were obtained from in vivo representative cases as described in Section 2.2. Moving scatterers were used to simulate the sequence of scanning US RF signals for one cardiac cycle (Stage 3). At the fourth stage, US RF signals for consecutive cardiac cycles could be composed, noise and artifacts optionally could be added thus approaching a more realistic model for the application of chosen signal processing algorithms. Finally, processing of simulated US RF signals, motion estimation, comparison of motion detection results with artery and surrounding tissue pulsation data set in Stage 2 as a reference, and relative errors of motion detection were estimated.

### 2.1. 3D Static Artery Model

The artery was modeled by a 3D map of scatterers, which is defined by scatterer number, positions, and scattering strength. A scattering map of the 3D artery model was generated according to the histological structure of the CCA [24]. The geometry and structure of the artery model are shown in Figure 2 and consists of three layers: *tunica intima, tunica media,* and *tunica adventitia.*

The artery was modeled as three concentric pipes, which correspond to three layers of the artery wall, keeping geometric dimensions of layers close to the dimensions of the in vivo CCA. According to our observation [25], CCA diameters range from 4.8 up to 7 mm; similarly, Krejza et al.’s research [26] showed CCA lumen dimensions to be 6.10 ± 0.80 mm. For illustrative example, dimensions were chosen to match in vivo recordings of a 23 year old male from our previous study [25]: the lumen diameter was *D*_0_ = 5 mm; the thickness of the intima–media complex was 0.7 mm; the thickness of the adventitia layer was 0.3 mm; the artery center was 17.5 mm deep from the surface (see Figure 2).

The first step in simulation is to define the media in which scattering structures will be embedded and, finally, digitally scanned with the modeled linear array transducer. Scatterers were randomly distributed within the volume of the artery model. The number of scatterers within the volume of the artery model is defined according to a corresponding resolution cell of the modeled linear array transducer [27]. A fully developed speckle pattern of a simulated ultrasound B-scan image would be achieved if there were 10 scatterers per resolution cell [28]. Therefore, we used this constant in the calculation of the scatterer’s number in our artery model. The scattering strength of the scatterers followed a Gaussian distribution with mean set to zero and the variance set according to the backscattering cross-section of the particular tissue [29], i.e., from the real in vivo B-scan image of CCA (see Figure 3). The amplitude of the B-scan image, which was normalized from 0 to 255 and represents the brightness of each pixel, was used to scale the variance.

We attributed a scattering strength for each scatterer according to its coordinates, e.g., if the scatterer belonged to the *intima* layer, its scattering strength was determined by the variance of the *intima* layer. Finally, we had a 3D scattering map of the 3D artery model (see Figure 4).

### 2.2. Simulation of Artery and Tissue Pulsation

The main purpose was to model not a single B-mode ultrasound image but to simulate a sequence of B-mode images that would reflect radial and longitudinal motions of the artery and surrounding tissue. Since the simulation of a single ultrasound RF frame in the Field II program can take up to several hours (depending on the volume of the model and the number of scatterers), cardiac cycles are similar and quasi-periodic with heart rhythm, and it was decided to model only one cardiac cycle of complex (i.e., longitudinal and radial) motion.

Radial and longitudinal motions of CCA needed for simulation were evaluated by ultrasound examination methodology, which are presented in [25] together with a clear description of the study population. Aiming to simulate surrogate structures, only three typical cases (out of 33 healthy volunteers) representing types of different waveforms of radial and longitudinal motions were selected from our own previous study [25]. The criteria for selection were the in vivo waveform morphologies. Types of morphologies were assumed according to publication on expert consensus review [15]. The CAROLAB software was used to estimate in vivo longitudinal and radial motions of CCA [30,31,32]. A region of interest (ROI) containing a well-contrasted speckle pattern of the distal vessel wall for longitudinal and radial motion evaluation was chosen in the first B-mode sequence frame. A kernel of the ROI was selected manually, with a size 3 × 0.5 mm. Estimated signals of longitudinal and radial motions were saved for further post-processing in MATLAB.

Motion signals were then filtered with a band-pass IIR filter (f_pass-lower_ = 0.5 Hz and f_pass-higer_ = 8 Hz) and then detrended, subtracting the mean of the resulting signal (see Figure 5). After this, three consequent heart cycles were selected manually in time. As mentioned before, only one heart cycle (see the green shadowed area in Figure 5) was used in motion modeling. In vivo data from our own previous study [25] were also used to determine the scatters’ strength, as shown in Figure 3, and also used as input in Stage 2 (see Figure 6). 

For simulations, the amplitude of the preselected periods of longitudinal and radial motion signals was normalized taking peak-to-peak amplitude in three levels: 0.5, 0.75, and 1 mm. The time axis was not normalized.

Mechanical deformation of the arterial wall consisting of two motions was simulated in this stage: (1) radial motion and (2) simultaneous longitudinal motion.

Even though the scatterers are randomly distributed within the artery model volume, their coordinates and scattering strength (scatterers amplitudes) are available for the simulation algorithm as a static map. This static map is an input of the Stage 2 simulation where at each scatterer’s coordinates we added a definite magnitude (the increment motion signal for the next time moment) and nature (radial or longitudinal) displacement (see Figure 6). Motion shape was used from real in vivo data (see Figure 5) [29]. The scatterer’s strength for particular tissue was kept constant as assigned at the initiation of modeling.

Each step of the artery and surrounding tissue pulsation simulation is explained in detail below, following a functional diagram of Stage 2 in Figure 6.

**Step 1.** Initially, the artery center of the artery model was set at 17.5 mm in depth (see Figure 2). Before adding radial and longitudinal displacement to a particular scatterer, the coordinate system was centered at the artery’s center (coordinates (0; 0) are at the artery center). In the next steps, Cartesian and polar coordinates were used for longitudinal and radial scatterers’ movement simulation accordingly;

**Step 2.** Simulating the radial motion component, before adding displacement to a particular scatterer, the Y and Z coordinates of the scatterers were recalculated from Cartesian to polar. Scatterers not belonging in a static map to the artery lumen (diameter *D*_0_ = 5 mm) were then selected and radial displacement *d_radial_*(*f*_1_) in the polar coordinate system was applied to each scatterer, where *f*_1_ was the first frame (time sample). Since the radial motion of the artery in the sequence of B-mode images was the sum of the motions of the proximal and distal artery walls (the motion of the proximal wall was equal to the motion of the distal wall), only the half of the displacement of the radial motion was added in the artery model. The magnitude of the displacement of artery layers and surrounding tissues depended on the coordinates of the scatterers, i.e., the farther the scatterer was from the lumen radius, the lower the magnitude of the displacement *d_radial_*(*f*_1_) added:(1)$$Slope_{radial}=\left\{\begin{array}{cc} 1, & y=r_{l}\\ e^{-b_{1}\left|y-r_{l}\right|}, & y>r_{l} \end{array}\right.,$$
where *y* is the radial position as the absolute radial distance from the artery axis, *r_l_* = 2.5 mm is the initial lumen radius, *b*_1_ = 0.17 expresses the slope of the radial displacement and was chosen the same as proposed by Stoitsis et al. [23]. In this way, the field of spatial-dependent radial displacements (see Figure 7) propagating away from the artery lumen radially and declining in surrounding tissues was modeled;

**Step 3.** After adding the radial displacement, the coordinates of the scatterers were again converted from polar to a Cartesian coordinate system.

Scatterers not belonging to the artery lumen in a static map (diameter *D*_0_ = 5 mm) were then selected, and longitudinal displacement d_longitudinal_(*f*_1_) in the X–Z plane was added to each scatterer. Other studies [33,34] have demonstrated that the *intima–media* complex shows a larger longitudinal motion than the *adventitia* layer. Therefore, for scatterers that were outside the *intima–media* complex (i.e., *adventitia* layer and surrounding tissues), longitudinal displacement was added with sloping according to distance from the artery center (see Figure 7):(2)$$Slope_{longitudinal}=\left\{\begin{array}{cc} 1, & r_{l}\leq y\leq r_{ma}\\ e^{-b_{2}\left|y-r_{ma}\right|}, & y>r_{ma} \end{array}\right.,$$
where *y* is the radial position as the absolute radial distance from the artery axis, *r_l_* = 2.5 mm is the lumen radius, *r_ma_* = 3.2 mm is the radius of the *media-adventitia* boundary, *b*_2_ = 1.08 expresses the slope of the longitudinal displacement and was chosen the same as proposed by Stoitsis et al. [23];

**Step 4.** In Step 2, the artery wall was displaced, the lumen diameter had changed; therefore, at each next frame of the sequence, the lumen was modeled as a cylinder with a variable diameter:*D_lumen_*(*f*_2_) = *D*_0_ + *d**_radial_*(*f*_1_),(3)
where *D_lumen_*(*f*_2_)—lumen diameter for the *2*nd frame of the sequence, *D*_0_—lumen diameter (const.), *d_radial_(f*_1_)—radial displacement between the two artery walls at the 1st frame of the sequence;

**Step 5.** The coordinate system’s center was returned to its initial position, i.e., at 17.5 mm in depth.

Thus, in each subsequent frame (*f*_2_*, f*_3_*,* etc.) we recalculated the coordinates of the scatterers in the artery and modeled lumen with a varying diameter model which yielded all remaining frames of artery motion.

### 2.3. Simulation of Sequential US Data

Using Field II software [35,36], the RF ultrasound signals were simulated from a time sequence of scattering 3D maps of the artery and tissues model. While simulating RF signals, a model of linear array transducer was used. Field II parameters used to simulate scanning with a linear array transducer are listed in Table 1. From the simulated US RF signals, the sequences of B-scan images were calculated. 

The resulting simulated B-scan image, in this case, consisted of 160 echo lines, the distance between two adjacent echo lines was 0.0625 mm, and the image width was 10 mm. One cardiac cycle consisted of 45 image frames. Several quasi-periodic cycles were compiled from the first one, while analysis of the one cycle peculiarities was the main concern in this paper.

To have the same formation of the B-scan image in both physical and digital scanning, we chose to load registered (i.e., physical scanning) and modeled (i.e., digital scanning) beamformed US RF signals into MATLAB. Then B-scan formation consisted of the following steps: (1) RF signals envelope detection; (2) log-compression from 0 to −73 dB. As an example, modeled B- scan image sequence is presented as Appendix A.

### 2.4. RF US Signal Processing Algorithms

From the simulated US RF data, patterns of radial and longitudinal motion could be evaluated using the chosen signal processing algorithms. We employed two algorithms: one to estimate only radial motion, and one to estimate both radial and longitudinal motions.

When we are interested in tissue properties based on motion only in one direction, a classical 1D cross-correlation (1DCC) is an option [37,38]. To obtain radial motion, the integral of inter-frame radial motion must be calculated.

Both radial and longitudinal tissue motions were evaluated by the OpenOpticalFlow (OOF) algorithm [39]. The simulated RF signals were normalized to 1, the envelopes (Hilbert transformation) of simulated RF signals were presented in decibels, the resulting data stronger than −90 dB were normalized from 0 to 255 and then passed as an input for OOF to identify motion between adjacent frames. Parameters used for OOF: size filter 4, Horn–Schunck estimator for an initial field *λ*_1_ = 120, and Liu–Shen estimator for refined estimation *λ*_2_ = 3000; these parameters were selected as the optimal combination after preliminary investigation within size filter values from 1 to 30, λ_1_ values from 1 to 1000, and λ_2_ from 100 to 10,000 by comparing the results with theoretical scatters movement and evaluating movement precision and smoothness.

The inter-frame displacements were later converted into the accumulated displacements relative to the first frame by respecting the fact that moving scatters change their coordinates.

The accuracy of the motion detection was evaluated in the midline (i.e., a middle scanning line) for various axial positions separately. The errors of radial and longitudinal motion detection were calculated by subtracting the theoretically defined motion pattern from the estimated motion. The root mean square (RMS) of these errors over time was normalized by the standard deviation of the theoretically expected displacements and expressed as normalized RMS error (NRMSE).

## 3. Results

To compare in vivo US data and in silico model data in more detail, they were presented in three different ultrasound modes: A, B, and M. Comparing the experimental and modeled data, it was found that in the A-mode the echoes from the lumen-intima and media-adventitia model boundaries are clearly visible in both cases (see Figure 8, A-mode subplots). In B-mode cases, the model walls are visible as a double-line pattern. Thus, the 3D model represents a structure similar to in vivo arterial walls (see Figure 8a,b).

For simulations, the amplitude of the preselected period of longitudinal and radial motion signals (see Figure 5c) was normalized taking peak-to-peak amplitude. In the example below (see Figure 8b), the radial and longitudinal motions’ peak-to-peak amplitude was 1 mm. In the in vivo case, the radial and longitudinal motions’ peak-to-peak amplitude of the preselected period (see Figure 5c) did not change and were 827 and 982 µm, accordingly, and the time axis was not normalized in this case. In M-mode, we can see the model movement according to the radial motion signal (see Figure 5c and Figure 8b). Both the proximal and distal model walls moved according to the same pattern, only in the opposite direction (see Figure 8b), and the motion amplitude of the distal and proximal walls was radial motion divided by two as simulated. In the in vivo case, a not so uniform motion pattern of both walls is seen (see Figure 8a). Likely, the ultrasound transducer was not perfectly positioned and was pressed into the neck during recording, which prevented the proximal wall of the artery from moving in the same pattern as the distal one. Another possible reason could be physiological that in vivo has been observed in animals [38,39] as asymmetric radial expansion and contraction of carotid artery.

In Figure 8b, the M-mode image indicates extreme displacement of the artery model at time instances *t*_1_ = 0 s and *t*_2_
*=* 0.15 s (frame rate was 52 frames per second). This extreme displacement was a 1000 µm motion difference between the proximal and distal walls. A detailed analysis of RF signals at the M-mode time instances *t*_1_ and *t*_2_ show a clear time shift of signals (see Figure 9).

A **radial** displacement of scatterers in the 3D model was also seen from the simulated US RF signals in Figure 9 and could be evaluated. Calculation of the displacement between M-mode instances *t*_1_ and *t*_2_ at distances 3.91 and 5.1 mm away from the simulated artery model center gave values of 400 and 330 µm, correspondingly (see Figure 9). For comparison, the displacements of scatterers in the model at those depths were 390 and 320 µm, correspondingly, what shows a good agreement of modeled displacement and calculated from the US RF signal. Displacements of tissue modeling scatterers were evaluated taking the half radial motion 500 µm between these time instances (see Figure 5c) and the theoretical radial motion decrement 0.78 and 0.64 at selected depths accordingly (see Figure 7).

Further analysis of radial motion showed that the movements estimated from simulated (as from B-mode in Figure 8b) US RF scanning frames matched the initially defined motion pattern used for modeling. For 1DCC and OOF algorithms comparison, their results are presented in Figure 10. The estimated motion waveforms had a similar radial motion pattern, amplitude, and decrement into the model tissue as the theoretically predefined motion in the dynamic 3D artery model (see Figure 10). Here, it is important to note that the estimated motion depends on the maximal peak-to-peak amplitude in the motion model and on the estimation algorithm applied. Spatial mapping of the radial displacement magnitude is provided in the Figure 11a,d sub-plots. Comparison of detected displacement waveforms after 1DCC and OOF algorithms was made in a point-wise manner that showed up to a 20 µm discrepancy for displacements of 0.5 mm peak-to-peak (see difference between dashed and colored waveforms in Figure 11b,e). With 1DCC detecting interframe motion only up to half a wavelength, similar output from the OOF algorithm was obtained. In this case, axial errors of detected displacements were of a similar level (see Figure 11c,f). Discrepancies in waveforms are provided in NRMSE form, which were up to a 0.5 level for both algorithms when model RF data had a maximal displacement peak-to-peak amplitude of 0.5 mm. The detection outputs of both algorithms had evident discrepancies when the artery wall in the model had displacements with maximal peak-to-peak amplitudes of 0.75 and 1 mm. In these modeled cases, the interframe motion exceeded half a wavelength of ultrasound, so the 1DCC algorithm, which is prone to ambiguities, was resisting the output. Therefore, non-detected motions in the models with maximal peak-to-peak amplitudes of 0.75 and 1 mm are indicated with wider white gaps in Figure 12a. At these spatial locations, NRMSE for radial errors between theoretically defined and estimated motion is not available. In other space locations with smaller magnitudes of model motion, the 1DCC tended to outperform the OOF algorithm. The OOF algorithm’s results often had an NRMSE larger than 0.3, while the 1DCC algorithm rarely had inaccuracies up to this level (see Figure 12a). Moreover, OOF was less able to detect motion at the edges of the scanning plane (compared in Figure 11a,d). However, in other cases, the results of both algorithms were comparable, and the estimated model motion differed from the theoretical one with median NRMSE between 0.11 and 0.22 regardless of the algorithm used and the peak-to-peak amplitude (see the medians in Figure 11c,f,i, Figure 12a and Figure 13 for more details). Therefore, simulate on errors cannot be attributed to the simulation method only. Motion differed from the theoretical one with median NRMSEs between 0.11 and 0.22 regardless of the algorithm used and the peak-to-peak amplitude (see medians in Figure 11c,f,i, Figure 12a and Figure 13). Therefore, simulation errors cannot be attributed to the simulation method only.

The same simulated sequences of US RF signals were used to estimate **longitudinal** model motions from the resulting B-scan frames using the OOF algorithm. Spatial mapping of longitudinal displacements magnitude is provided in Figure 11g. Detected displacement waveforms after the OOF algorithm are presented in Figure 11h, where detected displacement waveforms are at point-wise manner compared with theoretically defined. It can be noted, that detected displacements are in up to 110 µm discrepancy for theoretically defined displacements. Discrepancies of longitudinal waveforms are provided in NRMSE form (see Figure 11i), that are up to a 5 level, for both algorithms when model RF data were at a maximal displacement peak-to-peak amplitude of 0.5 mm. These indicate an approximately 10 times lower precision of longitudinal displacements representation in the model. The precision of longitudinal model motion estimation was lower: median NRMSE at the model artery wall ranged from 0.30 to 0.82 (see Figure 13), NRMSE at the model tissue become even worse at spatial points farther from the model artery wall where motion amplitudes were lower (see Figure 11i and Figure 12b with lateral NRMSEs).

Taken together with the radial displacement analysis, these results suggest that radial displacement amplitudes and morphology of their waveforms well represent the modeled motion. Longitudinal displacements, which rapidly decrease with the distance from the model artery, are represented also but with lower precision.

## 4. Discussion

In this study, we proposed a 3D artery model for simulation of radial and longitudinal motions of CCA and induced motions of surrounding tissues in silico.

The proposed algorithm, based on the distribution of moving scatterers, allows for building a digital phantom of a three-layer artery pulsating radially and longitudinally, surrounded by tissue experiencing induced descending motions. Radial and longitudinal motion time-patterns were set independently as well as decays of those motions in 3D space that enables the calculation of the model’s backscattered US RF signals. Such a dynamic model is useful for echoscopy simulations and serves as a reference for testing strain calculation algorithms, since all model parameters can easily be controlled.

The echoscopy simulation of the dynamic digital model by Field II showed that the artery scans presented by A-, B-, and M-mode images corresponded well to the movement models obtained from the pulsating artery wall in vivo cases. The evaluated simultaneous radial and longitudinal motion of the virtually scanned 3D structure over a single cardiac cycle was almost visually indistinguishable from generated motion pattern. The 1DCC and OOF algorithms showed acceptable estimated errors in radial movement.

A more detailed quantitative analysis was conducted using two US RF signal processing algorithms—1DCC and OOF—which have shown NRMSE values for radial and longitudinal movement patterns, also for different maximal peak-to-peak amplitudes and interpersonal differences of radial and longitudinal patterns. It showed discrepancies between the modeled movements of space-distributed scatterers representing artery walls together with surrounding tissues and movement estimates using RF signal cross-correlation and the optical flow method based on RF signal envelope tracking. Presented results show that estimated motion patterns are reproduced with NRMSE values from 0.11 to 0.22 for radial movement pattern and from 0.30 to 0.82 for longitudinal, which corresponds to the lateral scanning direction. It is interesting to notice that only one longitudinal movement was different from the others pattern for the 23 year old male (see Figure 5c), giving a higher NRMSE—0.82 (see Figure 13). This is evidence that the ability to evaluate longitudinal motion patterns is limited not only by lower lateral resolution, the performance of the OOF algorithm but also by the time dependence of the movement pattern. Lateral errors increase more rapidly with distance from the model artery wall (see Figure 11i), since movement decay is very steep and, consequently, the signal is too low for a good estimate. The 1DCC and OOF algorithms showed very close NRMSE values for radial movement patterns of the model’s artery wall, but the 1DCC algorithm failed with high amplitude movements (see Figure 12).

The results of the modeled radial and longitudinal displacement having a decay in the surrounding tissue away from the artery lumen or lumen-intima boundary, in some respects, went beyond previous reports [19,23] and show that the artery wall structure could be modeled in detail together with longitudinal and radial artery wall motion waveform and amplitude. The importance of longitudinal motion is pointed out in [15]; it was confirmed in clinical trials with carotid artery longitudinal motion using speckle tracking which was recently applied for diagnostics and monitoring [40]. The 3D finite element model of the patient-specific pulsating atherosclerotic carotid artery was used to simulate an ultrafast plane-wave echoscopy by Field II and to calculate errors of radial and circumferential strain estimates of the vessel wall [41]. Strain errors in this advanced mode of echoscopy were found to be approximately 2–4 percent, but the strain in the surrounding tissues and longitudinal strain errors was not analyzed. The outstanding publication on the 3D dynamic arteries model by S. Balocco et al. [42] included blood flow pulsation-induced vessel movements, Doppler simulations, and realistic vessel configurations. Estimated wall displacement calculated from M-mode Field II simulation error to compare with the FEM model was obtained from 3.92 to 10.6 percent, which is in the same range of errors as in the present article, which used less computational resources. The other application of Field II simulation together with modeling blood flow and pressure pulsation in a vessel was presented by Q. Zhang et al. [43]. The estimated error of wall displacement obtained from M-mode scans was up to 5.61–7.48, depending on the modeled stenosis level. Direct quantitative comparison of the results is problematic, since the estimation algorithms and modeling aims in available articles are different. Motion evaluation errors highly depend on the method and algorithm of evaluation, so modeling adequacy anyway is verified indirectly.

The limitation of this study comes from deficient knowledge about the in vivo slope of the longitudinal and radial motion deep into the tissue that are used as input of the model. For this moment, the slope is described by an exponential function, with parameters that were chosen according to a ratio of *b*_1_ and *b*_2_ by Stoitsis et al. [23]. Cinthio [37] evaluated such exponential function parameters from in vivo data but in a limited study and only from a healthy subject. For a more accurate evaluation of slope, the artery diameter could also be considered, since for illustrative purposes, we modeled CCA with a narrow diameter of 5 mm from those observed in the population [25,26], and it is expected that with a higher diameter, the slope will be lower. With aging, arterial stiffness increases [31], so motion slope should be evaluated not only in healthy but also in atherosclerotic patients to model different arterial stiffness. Tissue is complex anisotropic [44], viscoelastic nonlinear, nonhomogeneous media and those features were not modeled here.

The artery model, which uses both the shape and amplitude of the radial and longitudinal motion signals from real in vivo data, has not been proposed in previous studies. Further investigations are necessary to evaluate and confirm the authors’ results [23] of the spatial dependence of radial and longitudinal motions in vivo. The future directions of this study will focus on the development and validation of micro-motion tracking algorithms for the processing of RF signals or B-mode images.

## 5. Conclusions

A dynamic 3D model of a pulsating artery based on moving scatterers distributed in space was developed. Features of the model are realistic motion waveforms (radial and longitudinal) set independently as well as slopes of elastic motions deep into the tissues. Together with modeling of ultrasonic scanning with a linear array, this framework enables to simulate ultrasound RF signals backscattered from the scatterers that form the phantom of the artery and surrounding tissues and are related to the heterogeneous distribution of elasticity.

The algorithmic adaptable model enables evaluation of both the artery geometry (artery diameter, intima–media complex thickness) and the amplitude, shape, and slope of elastic motions deep into the tissues. This opens possibilities for the development and validation of algorithms for the processing of RF signals or B-mode images, e.g., *intima–media* complex segmentation, micro-motion tracking, elastography and, consequently, tissue characterization.

## Figures and Tables

**Figure 1 diagnostics-12-00232-f001:**
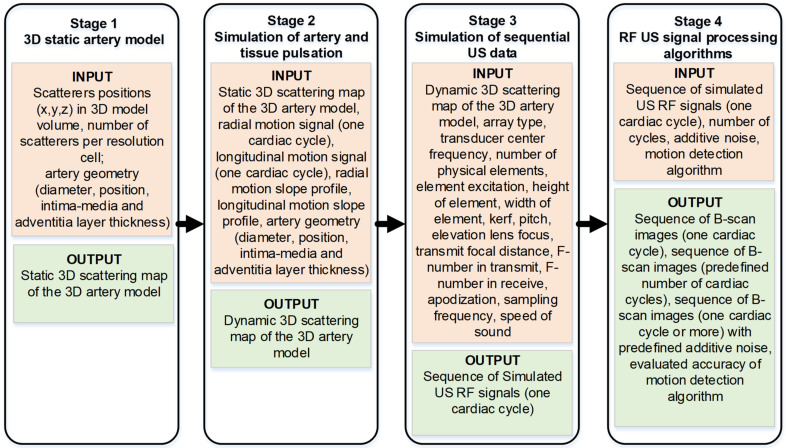
Overall simulation algorithm flow.

**Figure 2 diagnostics-12-00232-f002:**
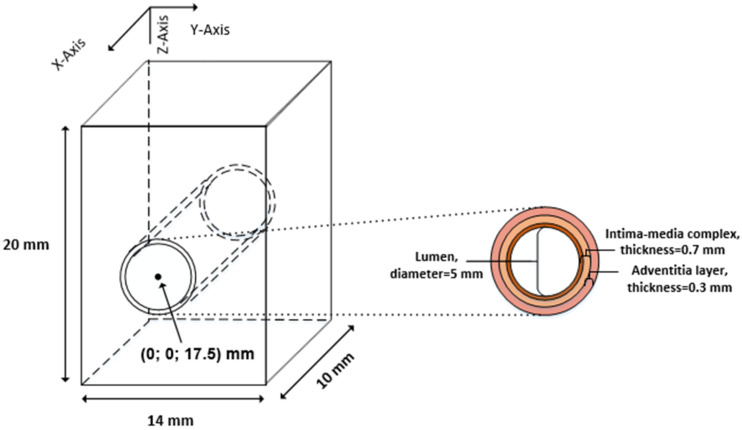
The general geometry of the artery model.

**Figure 3 diagnostics-12-00232-f003:**
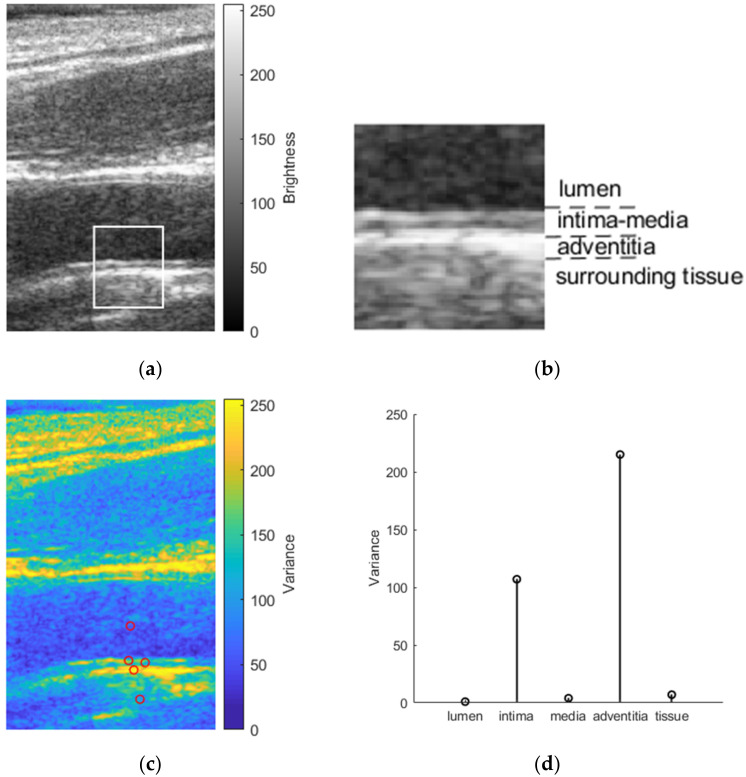
Determination of the scatterers’ scattering strength in the artery model: (**a**) B-scan image of the CCA; (**b**) detailed region (white rectangle in the B-scan image of CCA) of the distal wall with the lumen, the intima–media complex, the adventitia, and surrounding tissues; (**c**) variance map of CCA with preselected points from where variance values were selected to use in the artery model; (**d**) variance values assigned for a particular tissue of the artery model.

**Figure 4 diagnostics-12-00232-f004:**
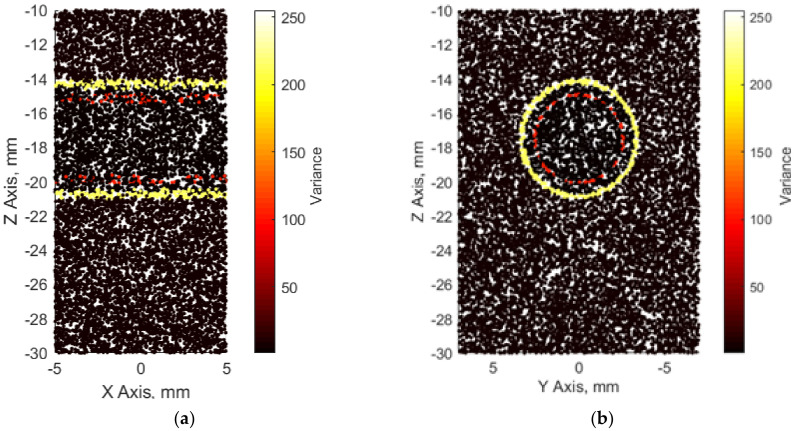
3D map of scatterers from different viewpoints representing the artery model: (**a**) longitudinal; (**b**) axial.

**Figure 5 diagnostics-12-00232-f005:**
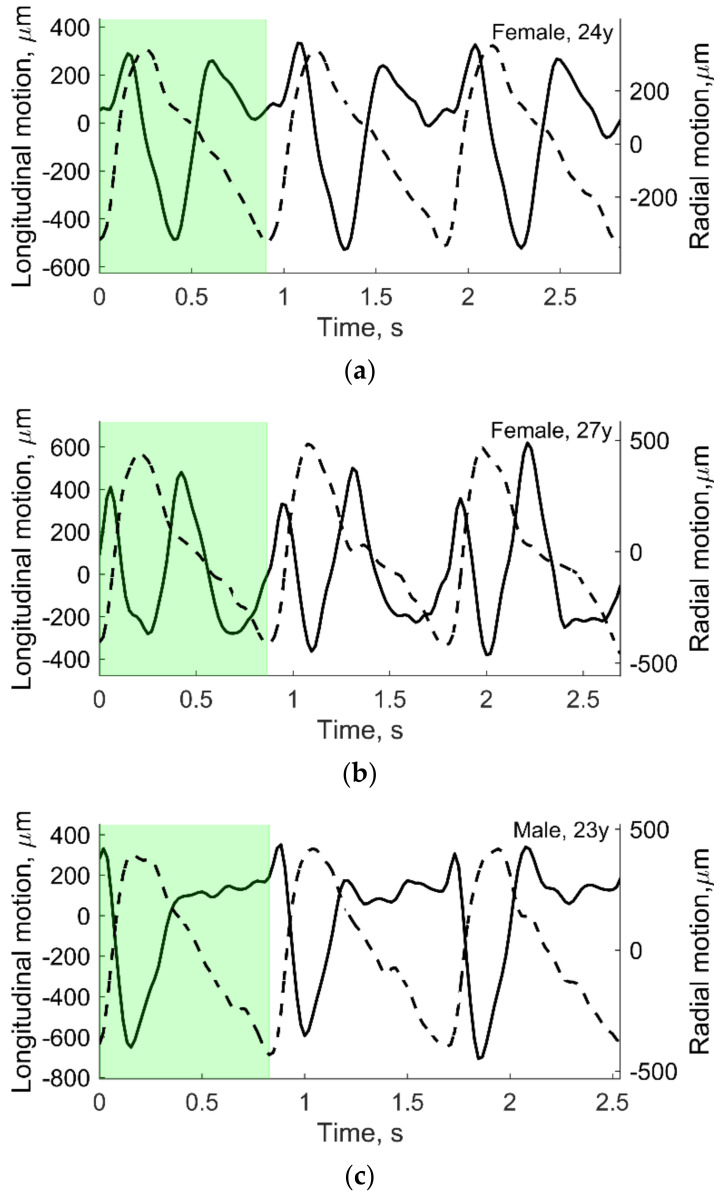
The processed in vivo radial (dashed line) and longitudinal (solid line) motions of CCA in three young subjects: (**a**) 24 year old female; (**b**) 27 year old female; (**c**) 23 year old male. Preselected periods (green shadowed area) were used to model artery and tissue motion. Adopted from [25].

**Figure 6 diagnostics-12-00232-f006:**
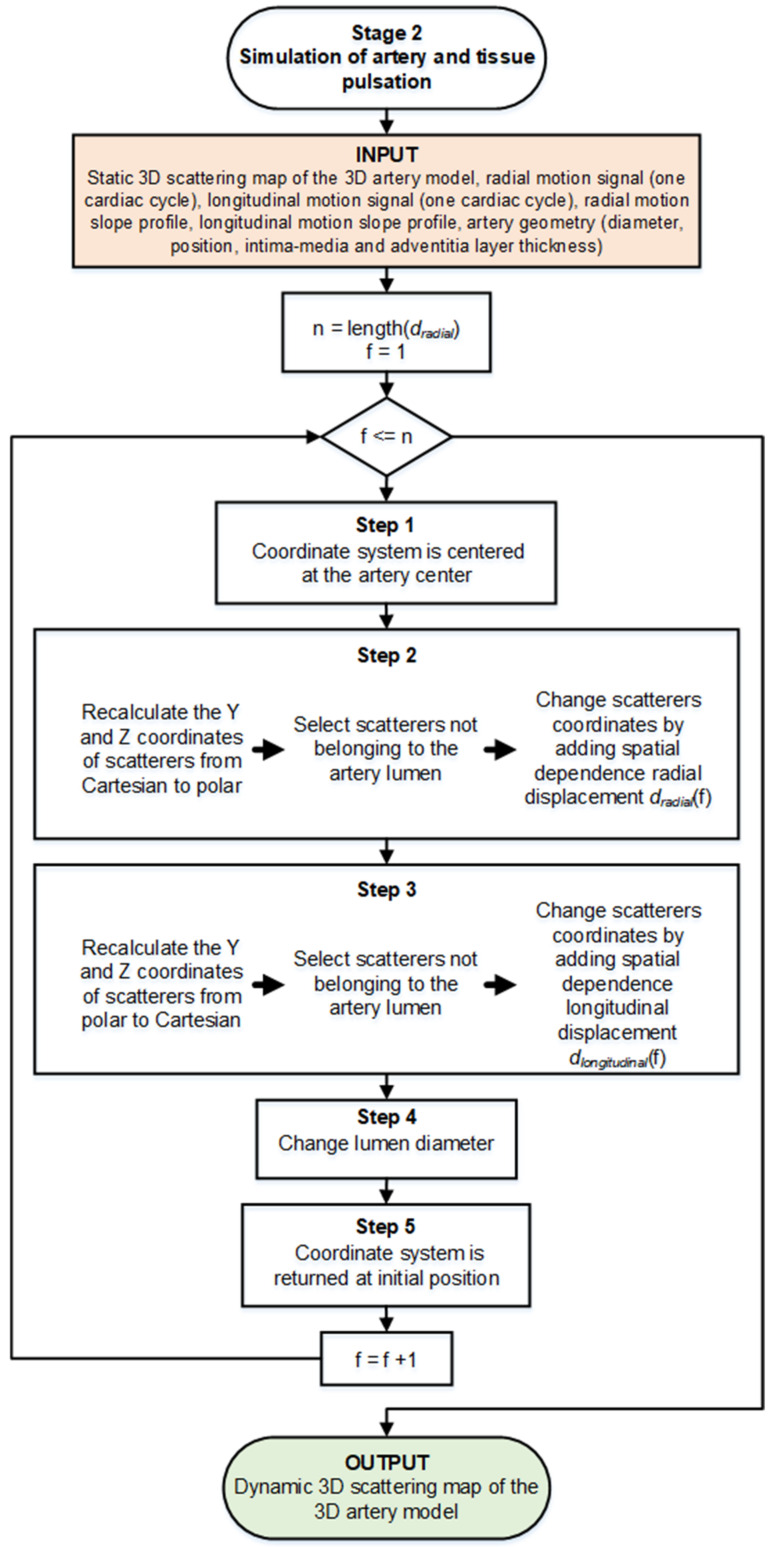
Functional diagram of Stage 2, simulation of the complex artery and tissue pulsation. Here, *f* corresponds to the currently analyzing frame number in the motion signal, and *n* corresponds to the total number of such frames.

**Figure 7 diagnostics-12-00232-f007:**
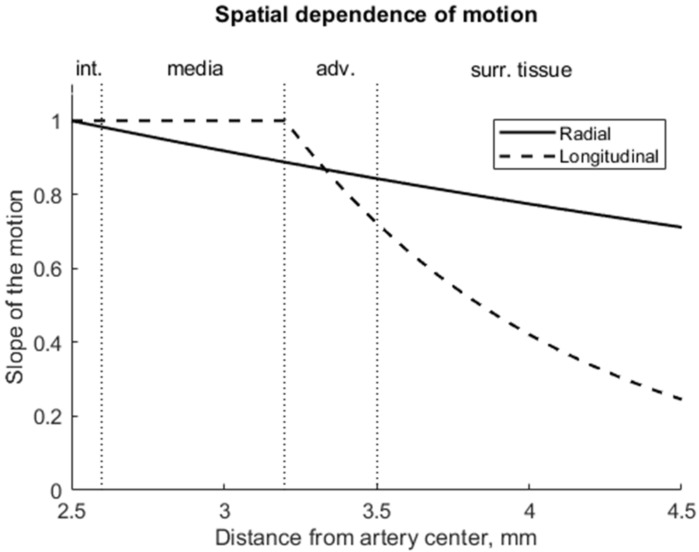
The amplitude of radial motion was greatest at the surface of the artery (2.5 mm) and decreased exponentially into the surrounding tissues; the amplitude of longitudinal motion was greatest at the *intima–media* complex (at 2.5–3.2 mm), decreased significantly into the adventitia layer (3.2–3.5 mm) and, finally, the longitudinal motion amplitude was negligible in the surrounding tissues.

**Figure 8 diagnostics-12-00232-f008:**
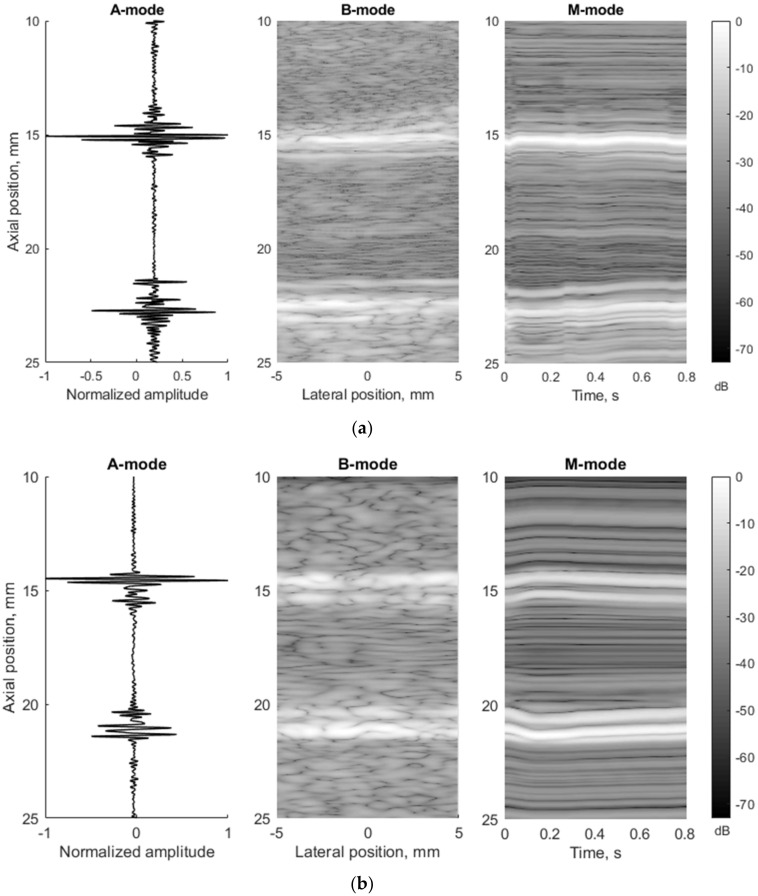
Comparison of RF signals form (A-mode, diastole), whole image (B-mode, diastole), and movement over time (M-mode) according to the artery (**a**) in vivo (23 year old male, motion amplitude was not normalized) and (**b**) in silico model data (23 year old male, motion amplitude was normalized to 1 mm).

**Figure 9 diagnostics-12-00232-f009:**
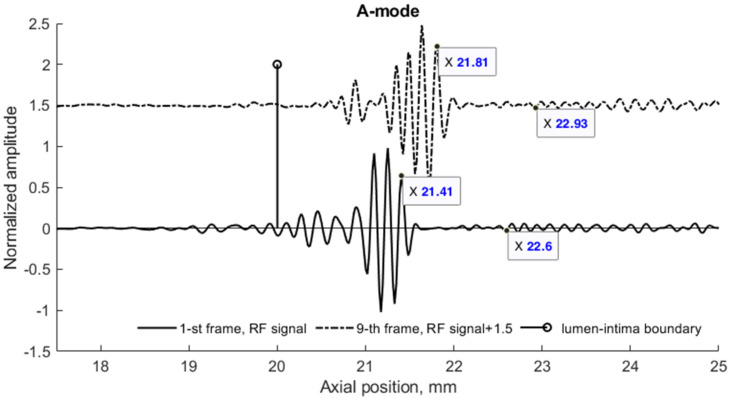
RF signals simulated by the 3D model of the pulsating artery at M-mode instances *t*_1_ and *t*_2_. Distal wall and surrounding tissues scattered RF signals (the artery center at 17.5 mm) depicting the radial wall displacement and its decrease in surrounding tissues. A 23 years old male, motion amplitude was normalized to 1 mm.

**Figure 10 diagnostics-12-00232-f010:**
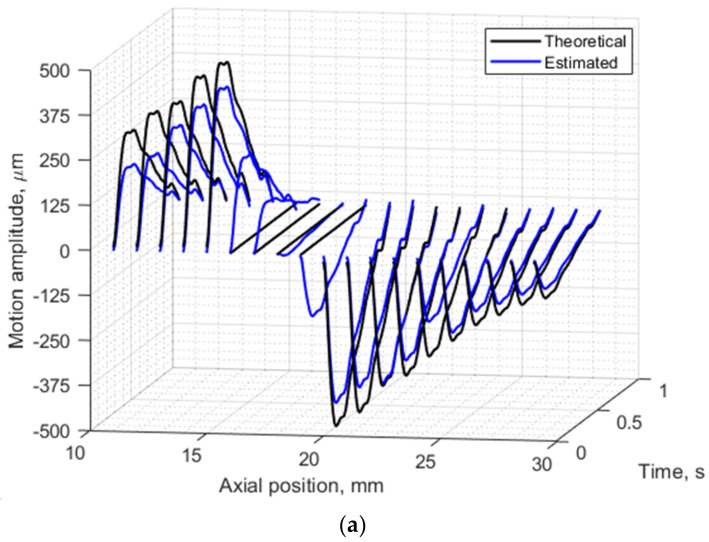
Radial motion of both the proximal and distal walls of the model artery and model tissue: (**a**) 1 mm peak-to-peak radial motion theoretically defined in 3D artery model and estimated from envelopes of simulated RF signals with OOF algorithm, (**b**) 0.5 mm peak-to-peak radial motion theoretically defined in 3D artery model and estimated from cross-correlation of simulated US RF signals. Segments were selected along the axial direction at lateral position 0 mm in adjacent frames of the modeled RF data. In total 19 axial positions from 11 to 29 mm along the axial direction. Theoretically defined waveform was from a 23 year old male.

**Figure 11 diagnostics-12-00232-f011:**
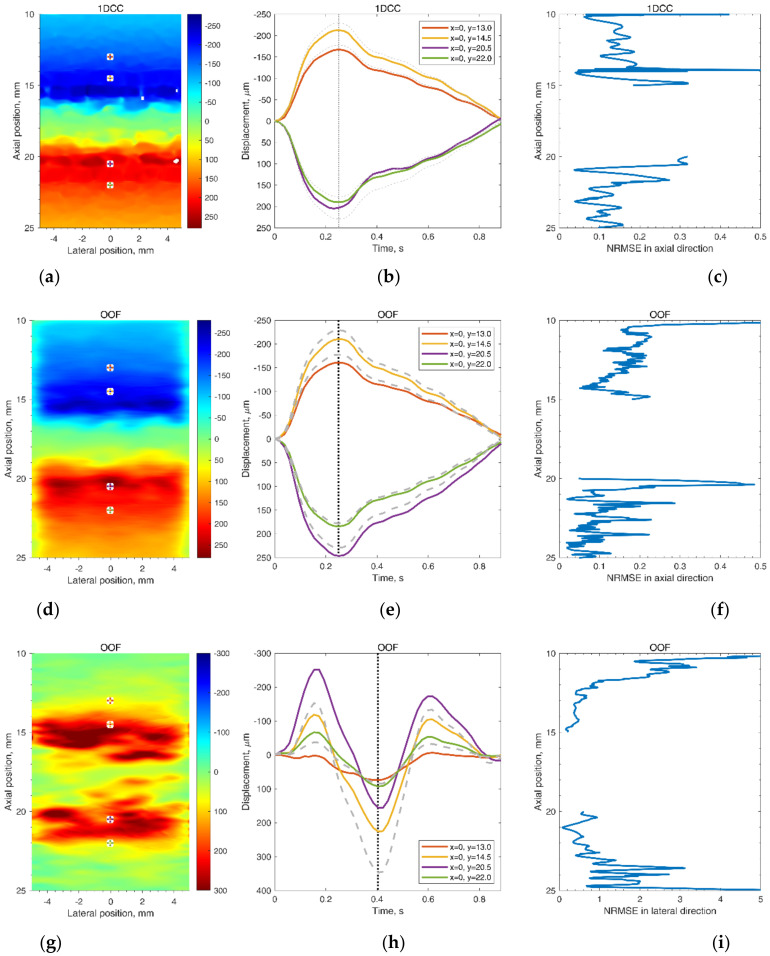
Frozen frames and selected point waveforms of radial (**a**,**b**,**d**,**e**) and longitudinal (**g**,**h**) movement in the scanning plane (**a**,**d**,**g**) in time at four selected points (**b**,**e**,**h**) and motion detection normalized root mean square error (NRMSE) in the midline of the scanning plane (**c**,**f**,**i**). Displacement magnitudes in the scanning plane (**a**,**d**,**g**) are at the moments (0.25 and 0.4 s) of maximal defined displacement that correspond to the vertical lines in the motion evolution over time in (**b**,**e**,**h**), where the grey, dashed lines depict the theoretical motion, and the solid, colorful lines depict 3D motion estimated by 1DCC (**b**) or OOF (**e**,**h**) algorithms (the displacement amplitudes were accumulated relative to the first frame). The waveform from the 24 year old female with a peak-to-peak amplitude normalized to 0.5 mm was used as the defined displacements waveform in the model.

**Figure 12 diagnostics-12-00232-f012:**
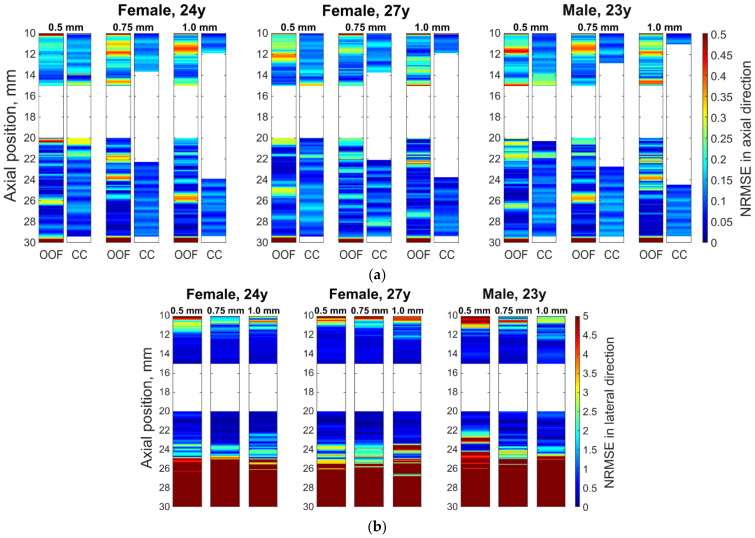
Normalized root mean square errors (NRMSEs) between theoretically defined and estimated motion in the midline of the scanning plane: (**a**) radial errors in the axial direction; (**b**) longitudinal errors in the lateral direction. (**a**,**b**) Sub-images: three blocks are the estimation errors for the model with three different defined waveforms (of 24 year old female, 27 year old female, and 23 year old male) with columns for maximal peak-to-peak amplitudes of 0.5, 0.75, and 1 mm, respectively. The motion was estimated in the axial direction by 1DCC and by OOF algorithm; motion in lateral direction by the OOF algorithm.

**Figure 13 diagnostics-12-00232-f013:**
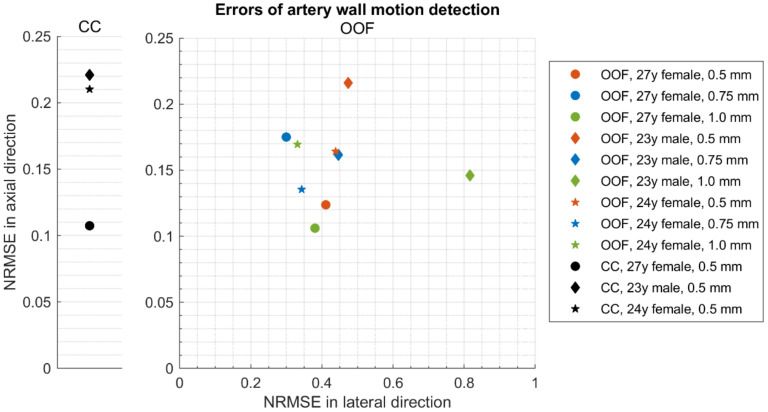
Medians of normalized root mean square errors (NRMSEs) between theoretically defined waveforms and estimated displacement waveforms in 3D motion models at axial positions from 14 to 15 mm and from 20 to 21 mm: 1DCC (see left part of the figure) and 2D OOF method (see right part of the figure).

**Table 1 diagnostics-12-00232-t001:** Parameters of virtual scanning with Field II.

Parameter	Value
Array type	Linear
Transducer center frequency	5 MHz
Number of physical elements	128
Element excitation	Hanning-modulated sinusoid of two cycles
Height of element	4 mm
Width of element	0.279 mm
Kerf	0.025 mm
Pitch	0.304 mm
Elevation lens focus	16 mm
Transmit focal distance	15.5 mm
F-number in transmit	3
F-number in receive	1.7
Apodization	Hanning
Sampling frequency	40 MHz
Speed of sound	1540 m/s

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
