# Peer review of "Simulation of Ultrasound RF Signals Backscattered from a 3D Model of Pulsating Artery Surrounded by Tissue"

_diagnostics, 2022, doi:10.3390/diagnostics12020232_

Round 1
Reviewer 1 Report
The authors have developed a computational dynamic 3D model of artery, based on human common coronary artery (CCA), able to reproduce the biomechanical behavior of the vessel submitted to pulsatile blood flow. The interest of this in silico model is that it can be used for the development and validation of algorithms of ultrasound elastography, a non-invasive method used to detect pathological alteration of arterial elasticity and stiffness.
General comment : the article is well written and the different methodological stages on the computational methodology are clearly presented. Validation of the model and its ability to reproduce in vivo ultrasound analysis is based on comparison of the prediction of the model with in vivo data obtained from human subjects. However, almost nothing is presented about these human subjects (according to the figure caption, one male (23y) and two females (24y and 27y)), as well as the method used for the in vivo measurement. I understand that in vivo recording s are not the core of the research. However, since the validation of the model by comparison with in vivo data is critical, precise information should be provided about the in vivo data.
Specific comments:
Line 127. The authors state that the tunica media is mainly composed of longitudinal smooth muscle cells. As far as I understand, cell orientation has not been integrated in the model. However, in CCA, smooth muscle cells are predominantly circularly, not longitudinally, oriented.
Line 138. The authors have based their model of CCA considering an internal diameter of 5 mm. However, human CCA internal diameter around 10 mm or more (see for example Hansen et al. BMC Neurology, 2021). How the authors have chosen the diameter used in their model (no reference is given)? What would be the influence of diameter change on the model predictions and their correspondence with in vivo data?
Author Response
Dear reviewer,
Thank you for your work in revising your paper and for giving insightful comments and suggestions for improving our manuscript.
Sincerely,
Authors of the manuscript

Reviewer 2 Report
This paper is to develop a dynamic 3D artery model reproducing the vessel motions in longitudinal and radial directions during a pulsatile cycle. Ultrasound imaging was simulated of linear scanning by Field II software to quantify mechanical micro-motions in the volume of the model. The results with reasonable errors would be helpful for the development and validation of elastography algorithms based on the motion detection. These results could be obtained from lots of works and research and be a basis to understand and evaluate the in vivo measurement of 3D dynamic vessel motion during a dynamic pulsating motion of the vessels. As the authors mentioned the limitations for this paper, the comparison with in vivo data are not presented and I’m wondering the reason since the in vivo measurements of ultrasound imaging is given. After using these as input data, it can also be used for evaluation even for a few cases. This is the most critical limitation of this paper. this paper needs to be re-edited for clarity and conciseness, since there are some redundancies and not well organized. Moreover, this paper needs to be re-edited for clarity and conciseness, since there are some redundancies and not well organized.
- The section of Materials and Methods can be shortened to remove redundancy.
- 1 can be reorganized to avoid the redundant phrase between input and output. The contents in inputs for stage 2 & 3 are much redundant in the figure and in the manuscript text.
- 1 also can be shortened for vessel layer description with a table or in the paragraph right after Fig. 2.
- 6, step 2 and 3 can be shortened to avoid redundant expressions and in page 9.
- 8 can be removed to describe in the text or combined with Fig. 9.
- Figures 12 ~15 need to be redrawn.
- Brightness and variance are not clearly described in Figure 3 and in the related paragraphs. Clarify these two terms.
- Are speckle patterns from 10 scatters per resolution for all volume in line 146 and Fig. 4? Especially lumen should be much smaller compared to vessel and the surrounding tissues. Then Fig. 9 & 10 are also better contrast. Or it may be required to control the grey scale since most of values are within -30 dB and the lower values were not much meaningful.
- How long it would take to get static and dynamic 3D scattering map and sequences of simulated USF RF signals? Please provide the computation time. The reason for one cycle simulation and the symmetric motion in Fig. 9 would be from the limitations of simulation time, I believe. So please have a table of computation time and load for each stage.
- The vessel from in vivo in Fig. 9 are expanding asymmetrically and in other papers. Please discuss about the asymmetrical expansion of the vessel.
- The signals stronger than -90 dB was normalized from 0 to 255. How did you do this? Linear scan conversion? Or exponential? Please describe the details.
- Describe the reason to normalize to 1 mm for simulated one because in vivo one is not but close to 1 mm, 827 and 982 um. If there is not measured one, how do you normalize amplitude?
- From figure 12 to 15, there are not much explained in the text. I think it’s better decreasing the materials and method, and add the details with quantitative comparison from the figures, or add tables for quantitative evaluation.
Line 119, D should be a typo.
Figure 5. (a), (b), (c) are not explained in the caption.
Line 283, deciBels
Line 305~6, rewrite the sentence.
Line 307-8, This would be shown in Method section.
Line 404, rewrite the sentence
Line 327, 1000 um à 1 mm, since Figure 10 caption is 1 mm for consistency and there is no reason to use um unit. How did you get this 1000 um? Averaged? Approximation?
Figure 11 caption should include that frame 9 is 0.15s later after frame 1 .
Line 347, radial is bolded and correct it.
Figure 11 &12 Time is not shown and the 3D figures are not much helpful to compare and read it.
Line 372, longitudinal is bolded and correct it.
Line 404, rewrite the sentence
Author Response

(The authors gave the same response as above.)

Round 2
Reviewer 1 Report
I thank the authors for their responses. However, I still consider problematic and unsatisfactory the absence of information regarding the obtention of the in vivo data on human beings of which the model is built. In the cited reference (Applied Sciences 2019, 9, 465), the study population was 33 healthy volunteers (9 male and 24 females). It is unclear whether the data presented in the present article are recordings already included in the previous publication, or new recordings. If the data are part of already published data, this should be precised, as well as why only 3 subjects among 33, and how they were selected. If the data have been obtained by new recordings, this corresponds to new in vivo experiments and it should be precised in the paper with the relevant information, including how the volunteers where chosen and if the ethical agreement obtained for the previously published paper also concerns the present study. So, the description of the obtention of in vivo recordings should be given more extensively in a specific paragraph.
Regarding the inner diameter, the authors have grounded their model on only one subject, and the diameter seems 2-fold less than the value found in the literature. Additionally, the authors have not answered my question regarding the influence of diameter change on the model predictions. In the discussion, the authors state the limitation of the study of Cinthia et al (ref. 36), done on one healthy subject, but their own study is limited also to just one healthy subject. The authors should discuss more accurately the question of the influence of the diameter on the model prediction, not only in the hypothesis of diameter reduction in the case of arterial stiffness, but also in the case of diameter values higher than that included in their model.
Author Response
Dear reviewer,
Thank you very much for the time you spend revising our manuscript. We sincerely appreciate all valuable comments and suggestions, which helped us to improve the quality of the article.
Sincerely,
Authors of the manuscript

Round 3
Reviewer 1 Report
The authors have answered adequately my remarks and modified their manuscript accordingly.